# ULD-Net: Enabling Ultra-Low-Degree Fully Polynomial Networks for Homomorphically Encrypted Inference

[*]**Xi Xie**
University of Connecticut
xi.xie@uconn.edu

[*]**Ran Ran**
North Carolina State University
rran@ncsu.edu

[*]**Jiahui Zhao**
University of Connecticut
jiahui.zhao@uconn.edu

**Bin Lei**
University of Minnesota - Twin Cities
lei00126@umn.edu

**Zhijie Jerry Shi**
University of Connecticut
zshi@uconn.edu

**Wujie Wen**
North Carolina State University
wwen2@ncsu.edu

**Caiwen Ding**
University of Minnesota - Twin Cities
dingc@umn.edu

## Abstract

Fully polynomial neural networks—models whose computations comprise only additions and multiplications—are attractive for privacy-preserving inference under homomorphic encryption (HE). Yet most prior systems obtain such models by post-hoc replacement of nonlinearities with high-degree or cascaded polynomials, which inflates HE cost and makes training numerically fragile and hard to scale.

We introduce ULD-Net, a training methodology that enables *ultra-low-degree* (multiplicative depth $\leq 3$ for each operator) fully polynomial networks to be trained from scratch at ImageNet and transformer scale while maintaining high accuracy. The key is a polynomial-only normalization, PolyNorm, coupled with a principled choice of normalization axis that keeps activations in a well-conditioned range across deep stacks of polynomial layers. Together with a special set of polynomial-aware operator replacements, such as polynomial activation functions and linear attention, ULD-Net delivers stable optimization without resorting to high-degree approximations.

Experimental results demonstrate that ULD-Net enables stable training of low-degree fully polynomial networks on large-scale model architectures and datasets. Applying ULD-Net to ViT-Small and ViT-Base achieves 76.70% and 75.20% top-1 accuracy on ImageNet, respectively, which are comparable to the original models and represent the first fully polynomial models successfully scaled to the ViT/ImageNet level. Additionally, ULD-Net outperforms several state-of-the-art open-source fully and partially polynomial approaches across diverse model architectures and datasets in both accuracy and HE inference latency.

The code is available at GitHub[†].

## 1 Introduction

Machine learning is increasingly delivered as a service (e.g., AWS SageMaker (ama), Azure ML (tea, 2016)), raising serious concerns regarding the confidentiality of user data and proprietary models. Homomorphic encryption (HE) (Dathathri et al., 2019; Kim et al., 2022) enables computation directly on ciphertexts, but today's deep networks rely on non-polynomial operators (e.g., ReLU, GELU, LayerNorm, Softmax) that are expensive or unsupported under HE. A popular

---

[*]These authors contributed equally.
[†]https://github.com/xiexi51/ULD-Net

workaround is to approximate such operators with high-degree polynomials or to offload them to alternative secure protocols (Tong et al., 2024; Lou et al., 2021; Ran et al., 2022). Unfortunately, high-degree or cascaded polynomials increase HE multiplicative depth and latency, and they remain brittle when scaled to large models and datasets.

**Our goal.** We revisit the problem from first principles: rather than approximating non-polynomial operators after training, can we directly train networks whose every layer is a low-degree polynomial, preserving accuracy while keeping HE cost small?

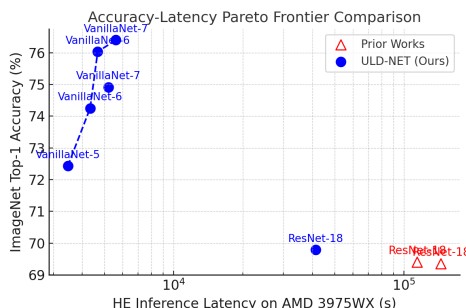

**This paper.** We present ULD-Net, a practical answer to that question. ULD-Net combines a new polynomial-only normalization layer, PolyNorm, with a specific set of training and architectural choices that together maintain tight control over activation ranges across depth. The design is agnostic to backbone (ResNets, VanillaNets, ViTs) and naturally HE-friendly: ultra-low-degree activations reduce ciphertext multiplications and multiplicative depth per layer, translating to faster encrypted inference.

Figure 1: ULD-Net achieves a better accuracy–latency Pareto frontier than prior SOTA fully polynomial works (Lee et al., 2021; Tong et al., 2024) on ImageNet.

**Why this is hard.** Polynomial functions of degree $\geq 2$ can explode outside a narrow input range; when stacked deeply, instability compounds and derails optimization, especially on large, high-variance datasets. Prior fully polynomial attempts that succeed at ImageNet often rely on high-degree or cascaded high-degree polynomials, trading stability for prohibitive HE cost (Hesamifard et al., 2017; Chou et al., 2018; Al Badawi et al., 2020; Garimella et al., 2021; Lee et al., 2021; Tong et al., 2024).

**Key ideas in ULD-Net.**

1. **Normalization-axis principle.** We show that choosing the normalization axis to match the geometry of polynomial layers (and the data layout) imposes effective range constraints with minimal overhead, improving stability at scale.

2. **PolyNorm: polynomial-only normalization.** PolyNorm implements strong numeric control using *only* additions and multiplications, making it natively compatible with HE while serving the same stabilizing role as common non-polynomial normalizers.

3. **End-to-end design recipe.** We provide an effective and broadly applicable design recipe—including suitable ultra-low-degree polynomial replacements for common activation, attention, and normalization operators, as well as auxiliary training techniques such as variance-aware penalty losses—that reliably trains fully polynomial networks on ImageNet and ViTs without high-degree approximations.

**Results at a glance.** ULD-Net-applied ResNet-18 achieves **69.79%** top-1 accuracy on ImageNet, outperforming the best fully polynomial baseline by +0.39% accuracy and **2.76×** HE inference speedup; and reaches 78.81% top-1 accuracy on CIFAR-100, surpassing the best partial polynomial baseline by up to +3.33% accuracy and 3.17× speedup. ULD-Net-applied ViT-Small is successfully trained on CIFAR-10 and Tiny-ImageNet, outperforming the best HE transformer baseline by up to +0.88% accuracy and 20.5× reduction in non-polynomial operator cost. Applying ULD-Net to the VanillaNet-5/6/7 family yields **72.43%/76.03%/76.40%** top-1 accuracy on ImageNet, with substantially lower HE latency than ResNet-18. Applying ULD-Net to ViT-Small/ViT-Base yields **76.70%/75.20%** ImageNet accuracy, representing, to the best of our knowledge, the first successful scaling of fully polynomial models to the ViT/ImageNet level. As shown in Figure 1, ULD-Net achieves a significantly better accuracy–latency Pareto frontier than prior works.

**Relation to prior work.** Partial replacement methods (Peng et al., 2023a) prune or relocate non-polynomial operators to reduce secure-inference overhead but still require costly non-polynomial handling. Fully polynomial approaches avoid that cost but often rely on high-degree

or cascaded polynomials to retain accuracy at scale (Lee et al., 2021; Tong et al., 2024). ULD-Net departs from both by training *ultra-low-degree* (multiplicative depth $\leq 3$) fully polynomial networks directly, enabled by PolyNorm and principled design choices that maintain numerical stability without sacrificing HE efficiency.

In summary, ULD-Net turns fully polynomial network design into a scalable, accuracy-preserving alternative for HE inference: it replaces post-hoc high-degree approximations with a pretraining-time solution that is simple to implement, architecture-agnostic, and demonstrably efficient under encryption.

## 2 RELATED WORKS

### 2.1 PARTIAL POLYNOMIAL REPLACEMENT

Partial replacement methods reduce, but do not eliminate, non-polynomial operators in deep networks (Mishra et al., 2020; Lou et al., 2021; Peng et al., 2023a). While attractive for ease of adoption, these approaches still require specialized handling (or offloading) of the remaining non-polynomial components during secure inference, especially in homomorphically encryption scheme, which leads to non-negligible latency and system complexity. More importantly, the numerical constraints for fully polynomial networks are substantially stricter than for partially replaced models. In practice, directly scaling partial-replacement techniques to the fully polynomial regime on large models or datasets often results in unstable training and/or impractical secure-inference costs.

### 2.2 FULLY POLYNOMIAL REPLACEMENT

A second line of work targets *fully* polynomial networks by approximating all non-polynomial operators with polynomials. Some prior works (Park et al., 2022; Aremu & Nandakumar, 2023; Ali et al., 2020; Garimella et al., 2021) are successful on small datasets but are unable to scale to ImageNet. Lee (Lee et al., 2021) proposed cascaded polynomials to reduce approximation error within a target range. Although cascading can reduce coefficient storage, the *effective* degree relevant to HE cost grows with the product of per-stage degrees, resulting in very high multiplicative depth on large-scale tasks; e.g., stable ImageNet training for ResNet-18 was reported to require an effective degree of 6075, which is prohibitive for HE inference. Tong et al. (Tong et al., 2024) reduced the cascaded degree to 81 via a suite of techniques (coefficient tuning, progressive approximation, alternating training, dynamic/static scaling), obtaining 69.4% top-1 on ImageNet; however, the training pipeline is complex and less portable across architectures, and we found it difficult to extend to fully polynomial ViTs. Diaa et al. (Diaa et al., 2024) use a quartic polynomial and introduce a penalty loss to constrain the inputs to the polynomial layers. However, it does not scale effectively to ImageNet and ViT-based architectures. Zimerman (Zimerman et al., 2024) reported fully polynomial ViT results on CIFAR-100, but did not specify the exact polynomial forms or degrees and did not release code, making the computational cost and stability trade-offs hard to assess.

### 2.3 POLYNOMIAL APPROXIMATIONS IN PRACTICAL HE INFERENCE

A complementary direction integrates polynomial approximation with system-level HE optimizations. For instance, NEXUS (Zhang et al., 2024) accelerate Transformer inference under HE by combining algorithmic changes with low-level HE engineering. Nevertheless, the approach still relies on iterative polynomial approximations for certain non-polynomial modules, which can require many steps and contribute substantially to multiplicative depth and runtime.

## 3 BACKGROUNDS

### 3.1 CKKS HOMOMORPHIC ENCRYPTION

Homomorphic encryption (HE) enables computation over encrypted data without decryption. Leveled HE (LHE) supports a bounded number of additions and multiplications, while Fully HE (FHE) allows unbounded computation via bootstrapping to refresh ciphertext noise (Gentry, 2009). The CKKS scheme (Cheon et al., 2017) is a widely used LHE scheme for approximate arithmetic on

fixed-point values encoded in complex slots, making it suitable for machine-learning workloads. CKKS supports ciphertext–ciphertext addition (Add), ciphertext–ciphertext multiplication (CMult), ciphertext–plaintext multiplication (PMult), and slot rotations via Galois automorphisms (Rotation, $\rho(ct, k)$). In typical implementations, CMult and Rotation are substantially more expensive than Add and PMult (e.g., up to $20\times$ slower), so overall latency is largely driven by the number of ciphertext multiplications, rotations, and the required multiplicative depth (Ran et al., 2023). Depth is controlled by modulus-chain management (rescaling) and determines whether bootstrapping is needed. Consequently, model designs that minimize polynomial degree and rotation usage are generally preferable for HE inference.

Notably, the *multiplicative depth* is generally regarded as the primary determinant of HE computation speed, and it grows proportionally with the logarithm of the polynomial degree.

## 3.2 FULLY POLYNOMIAL NETWORKS

Fully polynomial networks retain the original architecture while replacing every non-polynomial operation (e.g., ReLU, GELU, MaxPool, LayerNorm) with a polynomial operator, resulting in computation composed solely of additions and multiplications. This design eliminates the need for costly non-polynomial handling under HE and aligns with the operations that CKKS supports most efficiently. Common replacements include fixed or trainable low-degree polynomial activations (Lee et al., 2021; Peng et al., 2023b), AvgPool or polynomial pooling for MaxPool (Lee et al., 2021; Tong et al., 2024), and linear or iterative polynomial surrogates for normalization layers (Chen et al., 2022). Training is typically conducted in plaintext with the same polynomial operators that will be used at inference; any non-polynomial components used for stabilization must be removed or re-expressed before export to the encrypted setting. The degree of these polynomial operators directly impacts the multiplicative depth and number of ciphertext multiplications, and thus the practicality of HE inference. This motivates methods—such as ULD-Net —that achieve accuracy and stability with *ultra-low-degree* (multiplicative depth $\leq 3$) operators.

## 4 ULD-NET MODEL DESIGN

### 4.1 NUMERICAL CONSTRAINTS FOR FULLY POLYNOMIAL MODELS

Applying numerical constraints to the data flow of a fully polynomial model is primarily achieved through the normalization layers. The general form of a normalization layer is:

$$\text{Norm}[x] = \frac{x - \mathbb{E}[x]}{\sqrt{\text{Var}[x] + \epsilon}} \tag{1}$$

where $\mathbb{E}[x]$ and $\text{Var}[x]$ represent the mean and variance of the input $x$, respectively, and $\epsilon$ is a small value to prevent division by zero. The input tensor $x$ is constrained to a mean of zero and a variance of one over the chosen normalization axes. When a normalization layer is placed before a polynomial layer, this compression effect can effectively prevent the absolute value of the polynomial input from becoming excessively large, thereby avoiding divergent outputs. In particular, when the model contains many polynomial layers, it is generally necessary to insert a normalization layer before each polynomial layer to regulate the data flow and maintain stability.

### 4.2 CHOICE OF NORMALIZATION AXIS

We observe that different choices of normalization axes (i.e., the axes over which the statistics $\mathbb{E}[x]$ and $\text{Var}[x]$ are computed) result in different levels of stability for fully polynomial models. The most favorable choice for stability is to apply normalization to each sample in the batch, i.e., normalization over all axes except the batch axis. For example, in CNNs the tensor typically has the shape $[B, C, H, W]$ (samples, channels, height, width). Applying normalization over the $[C, H, W]$ axes provides the best stability for polynomial layers. For ViTs, the tensor typically has the shape $[B, N, D]$ (samples, patches, embeddings). In this case, normalization over the $[N, D]$ axes provides the most stable behavior.

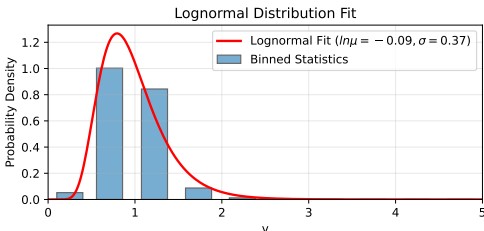

Figure 2: Experimental statistics of the $v$ values in the normalization layers of a deep neural network model (ResNet-18) during training on ImageNet, as defined in Eq. (6), and their lognormal distribution fit. The fitted parameters are $\ln\mu = -0.09$ (mean close to 1) and $\sigma = 0.37$.

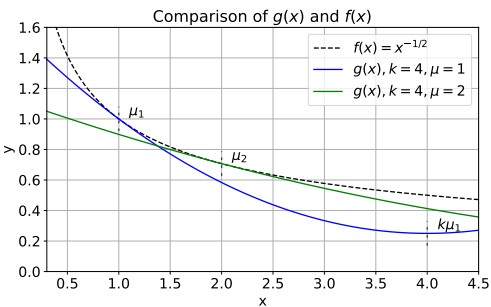

Figure 3: Comparison of $f(x) = \frac{1}{\sqrt{x}}$ and $g(x)$ with different $k, \mu$ settings. It shows that $g(x)$ fits $f(x)$ best around $\mu$ and remains monotonically decreasing within the interval $(0, k\mu)$ without exceeding $f(x)$.

To explain this phenomenon, we examine a model consisting of $n$ normalization layer and polynomial layer pairs. For simplicity, assume all polynomial layers share the same coefficients. For each sample $x \in \mathbb{R}^m$, the model computes the following sequence:

$$y = p_n(z_n(\cdots p_2(z_2(p_1(z_1(x)))) \cdots)) \tag{2}$$

where $z_1, \ldots, z_n$ are normalization layers, with the $i$-th layer having parameters $(\text{mean}_i, \text{var}_i)$, and $p_1, \ldots, p_n$ are all equal to a degree-$d$ polynomial $p(x) = \sum_{k=0}^{d} a_k x^k$, $d \geq 2$, $a_d \neq 0$. We will show through variance analysis that each sample requires its own suitable normalization layers parameters, otherwise numerical explosion may occur. Assume two samples $X$ and $X'$: $X \sim \mathcal{N}(\mu, \sigma^2 I)$ and $X' \sim \mathcal{N}(\mu, \sigma'^2 I)$. Suppose both adopt normalization layers parameters determined by $X$. Hence the first normalization layer uses $\text{mean} = \mu$, $\text{var} = \sigma^2$. For $X'$, the true standard deviation is $\sigma'$, and after the first normalization its variance becomes $v'_1 = \left(\frac{\sigma'}{\sigma}\right)^2 = r$. From the second normalization onward, the parameters are determined by $\text{mean}_i = \mathbb{E}[p(\mathbb{Z})]$, $\text{var}_i = \text{Var}[p(\mathbb{Z})]$, where $\mathbb{Z} \sim \mathcal{N}(0, I)$. Then, to compute the variance of $X'$ after the $(i+1)$-th normalization layer and polynomial layer, we approximate using the highest-order term of the polynomial:

$$v'_{i+1} \approx c\, v'^d_i, \quad c = \frac{a_d^2 \text{Var}[\mathbb{Z}^d]}{\text{Var}[p(\mathbb{Z})]} \tag{3}$$

Therefore, after $n$ layer pairs, the variance for $X'$ satisfies $v'_n \approx c^{\frac{d^n-1}{d-1}} r^{d^n}$. When $r > c^{-\frac{1}{d-1}} > 1$, we obtain $v'_n = O(r^{d^n})$, which shows an exponential-in-layer growth and an eventual explosion of variance. This demonstrates that if normalization are not computed separately for each sample, a fraction of samples may undergo numerical explosion. Moreover, the likelihood of such explosion increases with both the model depth and the number of samples, which verifies that fully polynomial models face a scalability challenge. Therefore, applying sample-specific normalization within each batch is the most favorable strategy for stability.

## 4.3 POLYNORM

However, in Eq. (1), aside from the two polynomial operations $\mathbb{E}[x]$ and $\text{Var}[x]$ (where $\text{Var}[x] = \mathbb{E}[x^2] - \mathbb{E}[x]^2$), there exists a non-polynomial operation:

$$f(x) = \frac{1}{\sqrt{x}} \tag{4}$$

Thus, we need to replace $f(x)$ with a polynomial function $g(x)$.

**Quadratic Approximation of $f(x)$.** We choose to approximate $f(x)$ with a quadratic function $g(x) = a(x-b)^2 + c$. To determine the values of $a$, $b$, and $c$, we proceed as follows:

Since the overall shape of $f(x)$ differs from that of the quadratic function, we focus on approximating $f(x)$ around a specified positive point $\mu$ and its neighborhood. We enforce that at $\mu$, $g(x)$ matches $f(x)$ in both function value and derivative value: $g(\mu) = f(\mu)$, $g'(\mu) = f'(\mu)$. In addition, noting that $g(x)$ is monotonically decreasing for $x \leq b$, we set $b = k\mu$, where $k$ can be specified within a certain range. The function $g(x)$ must open upwards and remain strictly positive, i.e., $a > 0$ and $c > 0$. By solving these conditions, we obtain:

$$a = -\frac{1}{4(1-k)\mu^{5/2}}, \quad c = \frac{5-k}{4\mu^{1/2}}, \quad k \in (1, 5). \tag{5}$$

Furthermore, we require that $g(x) \leq f(x)$ holds for all $x$ in the range $(0, k\mu)$. This ensures that $g(x)$ maintains good numerical constraint properties within this interval. It can be proven that this inequality holds if and only if it holds at $x = k\mu$. Simplifying the expression, we obtain $(5-k)\sqrt{k} \leq 4$, which leads to $k \geq 2.438$. Consequently, the range of $k$ is reduced to $[2.438, 5)$.

**The Expression of PolyNorm.** The function $g(x)$ has two key properties: it closely approximates $f(x)$ near $\mu$ and is monotonically decreasing within $(0, k\mu)$. To effectively utilize these properties, we cannot directly use $\text{Var}[x]$ as the input of $g(x)$. Instead, we first compute the relative value:

$$v = \frac{\text{Var}[x]}{\overline{\text{Var}}} \tag{6}$$

where $\overline{\text{Var}}$ is the historical average of $\text{Var}[x]$ during training, ensuring that $v$ has an expected value of 1 (as shown in Fig. 2, experimental statistics indicate that $v$ follows a lognormal distribution with a mean close to 1). Consequently, $\mu v$ has an expected value of $\mu$. Therefore, we use $\mu v$ as the input of $g(x)$ so that most inputs fall near the point where $g(x)$ exhibits its best properties. By combining Eq. (6) with Eq. (1) and ignoring $\epsilon$, we derive the following transformation:

$$\text{Norm}[x] = \frac{x - \mathbb{E}[x]}{\sqrt{\text{Var}[x]}} = \frac{x - \mathbb{E}[x]}{\sqrt{\frac{\mu \text{Var}[x]}{\overline{\text{Var}}}}} \cdot \sqrt{\frac{\mu}{\overline{\text{Var}}}} = (x - \mathbb{E}[x]) \cdot f(\mu v) \cdot \sqrt{\frac{\mu}{\overline{\text{Var}}}}$$

By replacing the function $f$ with the quadratic function $g$, we obtain the expression of PolyNorm:

$$\text{PolyNorm}[x] = (x - \mathbb{E}[x]) \cdot g(\mu v) \cdot \sqrt{\frac{\mu}{\overline{\text{Var}}}} \tag{7}$$

where

$$g(x) = -\frac{(x - k\mu)^2}{4(1-k)\mu^{5/2}} + \frac{5-k}{4\mu^{1/2}}, \quad k \in [2.438, 5) \tag{8}$$

Here, $k$ and $\mu$ are fixed parameters, and $\overline{\text{Var}}$ is a fixed value during inference. Consequently, $\sqrt{\frac{\mu}{\overline{\text{Var}}}}$ is also a precomputable fixed value. Thus, $\text{PolyNorm}[x]$ serves as the polynomial replacement for $\text{Norm}[x]$. We can also apply this replacement during both the training and inference phases in order to maintain greater consistency.

**The Numerical Constraint of PolyNorm.** PolyNorm constrains inputs with variance less than or equal to $k$ times the historical average to have zero mean and variance no greater than 1. The proof is provided in Appendix A.

**Analysis of Hyperparameters.** Considering the average value of $\frac{g(x)}{f(x)}$ over the interval $(0, k\mu)$, denoted as $R$, we have:

$$R = \frac{1}{k\mu} \int_0^{k\mu} \frac{g(x)}{f(x)} \, dx = \frac{4k^{5/2}}{105(k-1)} + \frac{(5-k)k^{1/2}}{6}$$

It can be proven that $R$ is monotonically decreasing for $k \in (2.438, 5)$ and satisfies $0.532 \leq R \leq 0.913$. On the other hand, we note that $v$ follows a lognormal distribution, as shown in Fig. 2. The lognormal distribution exhibits a long-tail characteristic. Experimental statistics show that the probability of $v$ exceeding 3 is still approximately $3 \times 10^{-4}$, while the probability of exceeding 5 falls below $1 \times 10^{-5}$. Since the monotonic decreasing range of $g(x)$ increases as $k$ increases, increasing the value of $k$ enhances the numerical stability of PolyNorm over a wider range of $v$

values. That is, although decreasing $k$ improves the fitting accuracy of $g(x)$ to $f(x)$, too small a value of $k$ will lead to a higher proportion of samples with numerical instability. Fig. 3 illustrates the comparison between $g(x)$ under different $k$ and $\mu$ values, and $f(x)$. We empirically verify that $k = 4$ is a choice that balances both fitting accuracy and stability. The value of $\mu$ has a relatively minor impact on accuracy. In this work, we consistently use the empirical hyperparameters $k = 4$ and $\mu = 2$. The corresponding $g(x)$ is:

$$g(x) = 0.01473x^2 - 0.23565x + 1.11937$$

### 4.4 OVERALL DESIGN RECIPE

In summary, we adopt the following fully polynomial replacement strategy:

For activation functions (e.g., ReLU, GELU), we employ trainable low-degree polynomial activations (Lee et al., 2021; Peng et al., 2023b), combined with dropout to reduce overfitting. This denoted as PolyAct, is defined as

$$\text{PolyAct}(x) = \text{Dropout}\left(\sum_{i=0}^{n} \alpha_i c_i x^i\right), \tag{9}$$

where $\alpha_i$ are trainable coefficients and $c_i$ are fixed adjustment factors. In this work, we consistently adopt ultra-low-degree polynomials with $i \leq 3$. For softmax attention replacement in ViT, we adopt Linear Attention with Rotary Position Embedding (RoPE) (Su et al., 2024), defined as

$$\text{LinearAttn}(x) = \text{RoPE}(Q) \cdot \text{RoPE}(K)^\top \cdot V, \tag{10}$$

where $\text{RoPE}(\cdot)$ denotes the rotary position embedding operation, which is composed entirely of runtime constants and linear operations. The max pooling layer is replaced with the average pooling layer (Gilad-Bachrach et al., 2016). Normalization layers are replaced with the proposed PolyNorm layer defined in Eq. (7). To provide better initialization for $\overline{\text{Var}}$ (the running average of $\text{Var}[x]$ during training), we still employ Eq. (1) during the warmup training epochs. To further improve the stability and accuracy of PolyNorm, when adopting Eq. (7) in training we introduce two penalty loss terms, $\mathcal{L}_1$ and $\mathcal{L}_2$, based on the magnitude of $v$ (see Eq. (6)) and its deviation from 1:

$$\mathcal{L}_1 = \frac{1}{N} \cdot \sum_{i=1}^{N} v_i \cdot \lambda_1, \quad \mathcal{L}_2 = \frac{1}{N} \cdot \sum_{i=1}^{N} (v_i - 1)^2 \cdot \lambda_2, \tag{11}$$

where $N$ is the number of PolyNorm layers, $v_i$ is the $v$ value of the $i$-th PolyNorm layer, and $\lambda_1, \lambda_2$ are scaling coefficients. The introduction of $\mathcal{L}_1$ suppresses excessively large $v$ values, enhancing stability, while $\mathcal{L}_2$ encourages the distribution of $v$ values to be closer to 1, which is the optimal region for the function $g(x)$.

Fig. 4 illustrates our overall design framework, which achieves near-extreme acceleration of the security scheme while largely preserving the capability of the original model.

## 5 EXPERIMENT

### 5.1 EXPERIMENTAL SETUP

**Architectures and Datasets.** We evaluate ULD-Net on both CNN models (ResNet (He et al., 2016), VanillaNet family (Chen et al., 2023)) and ViT-Small (Dosovitskiy et al., 2021; Wightman, 2019). The datasets used include CIFAR-10, CIFAR-100, Tiny-ImageNet, and ImageNet.

**Comparison with Prior Works.** We conduct a fair comparison of ULD-Net in terms of both model accuracy and HE inference speed against a diverse set of state-of-the-art approaches. These include fully polynomial replacement methods (Lee et al., 2021; Tong et al., 2024), partial polynomial replacement methods (Cho et al., 2022; Peng et al., 2023a), and recent Transformer HE inference acceleration work (Zhang et al., 2024).

**ULD-Net Setup and Training.** We follow the replacement strategy described in Subsection 4.4. Additional hyperparameters and training details are provided in Appendix B. Training is conducted using PyTorch 2.7 on 8 NVIDIA A100 GPUs.

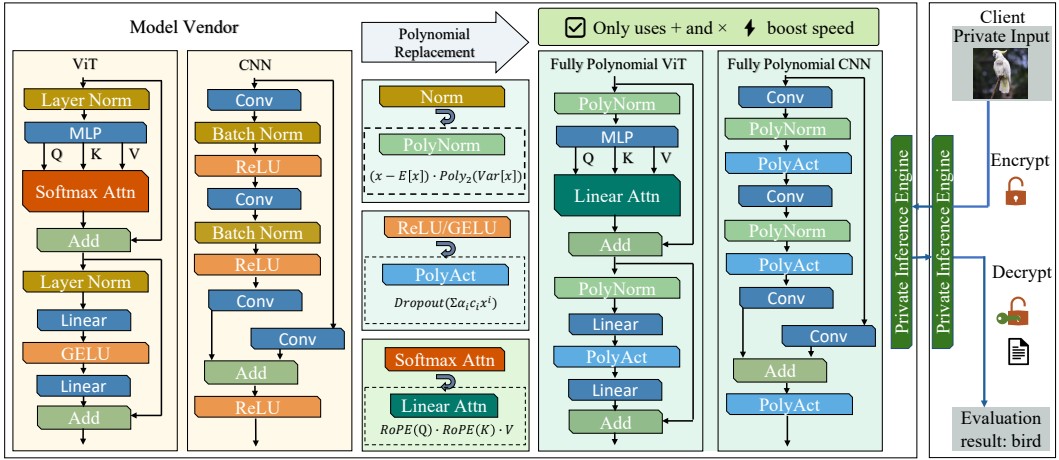

Figure 4: Our low-degree fully polynomial model design framework. $\text{Poly}_2(\cdot)$ denotes a quadratic polynomial function.

Table 1: Comparison with SOTA fully polynomial replacement methods in terms of accuracy and HE inference latency (on ResNet-18/ImageNet, original accuracy: 69.76%). The reported latency covers both a single polynomial activation function and the entire model.

| ResNet-18 / ImageNet (Fully Polynomial) | Activation Degree | Test Acc. | Activation Latency (s) | Speedup | Model Latency (s) | Speedup |
|---|---|---|---|---|---|---|
| Lee et al. (2021) | 6075 | 69.35% | 16448 | 16.06× | 144896 | 3.50× |
| SMART-PAF | 81 | 69.40% | 8311 | 8.12× | 114277 | 2.76× |
| ULD-Net (Ours) | 2 | 69.79% | 1024 | – | 41408 | – |

**HE Latency Evaluation.** We evaluate the HE inference latency of all experiments on a machine equipped with an AMD Threadripper 3975WX CPU under the single-thread setting. Microsoft SEAL version 3.4.5 (SEAL) is used to implement the RNS variant of the CKKS scheme (Cheon et al., 2018). We measure computation latency by running 20 samples and reporting the average. Our encryption parameter setting follows prior work (Tong et al., 2024), with polynomial degree $2^{15}$ and modulus 881, ensuring a 128-bit security level (Albrecht et al., 2021; 2015) against known LWE attacks.

## 5.2 EXPERIMENT RESULTS

**Comparison with SOTA Fully Polynomial Replacement Methods.** We compare ULD-Net with Lee (Lee et al., 2021) and SMART-PAF (Tong et al., 2024) on fully polynomial ResNet-18 training with ImageNet. The results are presented in Table 1. All three methods enable stable training of fully polynomial ResNet-18 on ImageNet and achieve test accuracy comparable to the original model, with the main difference lying in the polynomial degree. Lee and SMART-PAF approximate the activation function using cascaded polynomials, leading to very high equivalent polynomial degrees and correspondingly high HE multiplicative depth. The multiplicative depth is generally regarded as the primary determinant of HE computation speed, and it grows proportionally with the logarithm of the polynomial degree. In contrast, ULD-Net, with the aid of PolyNorm, performs the replacement entirely with quadratic activation functions (polynomial degree and multiplicative depth both equal to 2). This not only achieves an 8.12× speedup for the activation function and a 2.76× speedup for the entire model over SMART-PAF (the current SOTA), but also benefits from the non-linear functionality provided by quadratic activation functions (Peng et al., 2023b), leading to the highest accuracy among all methods (i.e., +0.39% higher than SMART-PAF and +0.03% higher than the original model).

**Comparison with SOTA Partial Polynomial Replacement Methods.** For the remaining ReLU functions in partial replacement methods, we adopt the approximation proposed by Lee et al. (2021). As shown in Table 2, ULD-Net demonstrates a significant advantage in HE latency compared with SNL (Cho et al., 2022) and AutoReP (Peng et al., 2023a), achieving up to a 2.88× speedup in activation latency and a 3.17× speedup in overall model latency. This improvement arises because full

Table 3: Comparison of accuracy and HE latency of non-polynomial operators for fully polynomial ViT-Small. Patch size is 4 on CIFAR-10 and 16 on Tiny-ImageNet.

| Dataset Original Acc. | Method | Test Acc. | Non-Polynomial Operator Latency (s) | | | | Speedup |
|---|---|---|---|---|---|---|---|
| | | | Softmax | LayerNorm | GELU | Total | |
| CIFAR-10 | NEXUS | 91.39% | 3055 | 2080 | 2860 | 7995 | 20.5× |
| 91.77% | ULD-Net (Ours) | 91.48% | 156 | 156 | 78 | 390 | – |
| Tiny-ImageNet | NEXUS | 60.52% | 9259 | 6304 | 8668 | 24231 | 20.5× |
| 60.90%[*] | ULD-Net (Ours) | 61.40% | 472 | 474 | 236 | 1182 | – |

[*] Using RoPE Attention.

replacement reduces the overall evaluation circuit depth, thereby lowering the need for bootstrapping and further decreasing the end-to-end model latency.

Table 2: Comparison with SOTA partial polynomial replacement methods (on ResNet-18/CIFAR-100, original accuracy: 77.84%).

| Method | ReLU Replace Ratio | Test Acc. | Activation Latency (s) | Model Latency (s) |
|---|---|---|---|---|
| SNL | 0.88 | 73.75% | 45 | 2052 |
| AutoReP | 0.87 | 75.48% | 46 | 2053 |
| AutoReP | 0.93 | 74.92% | 35 | 2042 |
| ULD-Net (Ours) | 1 | 78.81% | 16 | 647 |

Moreover, partial polynomial replacement methods generally cannot be directly extended to fully polynomial replacements, as they fail to provide the required numerical stability. From the accuracy perspective, ULD-Net fully leverages the benefits of polynomial activations (quadratic in this case), achieving accuracy even higher than the original model (+0.97%) and outperforming AutoReP by +3.33%. Therefore, ULD-Net clearly surpasses the existing SOTA partial polynomial replacement methods in both accuracy and efficiency.

**Evaluation with ViT-Small and ViT-Base.** Our experiments demonstrate that ULD-Net scales to ViT-Small and ViT-Base, successfully training both models on ImageNet and achieving accuracy comparable to the original versions (76.7% vs. 76.5% and 75.2% vs. 75.3%, reported by Heo et al. (2021)). In Table 3, we compare ULD-Net with the recent Transformer HE inference acceleration framework NEXUS (Zhang et al., 2024) on ViT-Small. Although NEXUS is based on polynomial approximation, it requires very high polynomial degrees: the multiplicative depths of Softmax, LayerNorm, and GELU reach 16, 16, and 14, respectively. In contrast, ULD-Net uses RoPE, PolyNorm, and PolyAct (cubic in this case), which require multiplicative depths of only 2, 3, and 3. This leads to a 20.5× speedup over NEXUS in total non-polynomial operator latency. ULD-Net also achieves strong accuracy: on CIFAR-10 it is +0.09% higher than NEXUS (-0.29% compared to the original model), and on Tiny-ImageNet it exceeds NEXUS by +0.88% and the original model by +0.50%.

These results demonstrate that ULD-Net has been validated at the scale of modern Transformer models up to ViT-Base (approximately 86M parameters). Extending the proposed low-degree polynomial design to even larger-scale Transformers, such as ViT-Large, BERT-Large, or GPT-2, constitutes an important direction for future work.

Table 4: Extended experiments of ULD-Net with the VanillaNet family on ImageNet.

| Model Original Acc. | Activation Degree | Test Acc. | Activation Latency (s) | Model Latency (s) |
|---|---|---|---|---|
| VanillaNet-5 72.49% | 2 | 72.43% | 478 | 3469 |
| VanillaNet-6 76.36% | 2 | 74.25% | 597 | 4337 |
| | 3 | 76.03% | 939 | 4678 |
| VanillaNet-7 77.98% | 2 | 74.91% | 717 | 5204 |
| | 3 | 76.40% | 1126 | 5614 |

**Experiments with the VanillaNet family.** VanillaNet (Chen et al., 2023) is a lightweight CNN model that achieves strong accuracy performance on ImageNet. We apply ULD-Net to VanillaNet-5/6/7 to validate its broad applicability and scalability. As shown in Table 4, ULD-Net successfully enables fully polynomial VanillaNets to be trained stably on ImageNet, achieving accuracy close to that of the original models

while maintaining substantially lower overall HE latency. In particular, VanillaNet-7 reaches 76.40% accuracy with a $7.4\times$ HE latency speedup compared to ResNet-18.

## 6 CONCLUSION

We introduced ULD-Net, a training methodology for ultra-low-degree fully polynomial networks at ImageNet and transformer scale. With polynomial-only normalization and operator replacements, ULD-Net overcomes prior instability and scalability issues. Experiments on ViT-Small and ViT-Base achieve 76.70% and 75.20% top-1 accuracy on ImageNet, representing the first ultra-low-degree fully polynomial ViT models trained at this scale.

## 7 ACKNOWLEDGMENTS

This research was supported in part by NSF ECCS-2534861, CNS-2505747, and CNS-2348733.

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

## A PROOF OF POLYNORM NUMERICAL CONSTRAINT

Considering the mean and variance of PolyNorm[$x$], it is evident that

$$\mathbb{E}[\text{PolyNorm}[x]] = \mathbb{E}[x - \mathbb{E}[x]] \cdot g(\mu v) \cdot \sqrt{\frac{\mu}{\text{Var}}} = 0$$

and then

$$\begin{aligned}
\text{Var}[\text{PolyNorm}[x]] &= \mathbb{E}[\text{PolyNorm}[x]^2] - \mathbb{E}[\text{PolyNorm}[x]]^2 = \mathbb{E}[\text{PolyNorm}[x]^2] \\
&= \mathbb{E}[(x - \mathbb{E}[x])^2] \cdot g(\mu v)^2 \cdot \frac{\mu}{\text{Var}} \\
&= \text{Var}[x] \cdot \frac{\mu}{\text{Var}} \cdot g(\mu v)^2 \\
&= \mu v \cdot g(\mu v)^2 = \left(\frac{g(\mu v)}{f(\mu v)}\right)^2 \leq 1 \quad \text{for } \mu v \in (0, k\mu]
\end{aligned}$$

Thus, PolyNorm ensures an expected value of 0 and a variance no greater than 1 for inputs satisfying $\text{Var}[x] \leq k \cdot \overline{\text{Var}}$.

## B ADDITIONAL TRAINING HYPERPARAMETERS AND SETTINGS

For additional implementation details, please refer to our GitHub repository and accompanying instructions, which serve as the authoritative reference for all experimental configurations and hyperparameter settings.

**Polynomial Operator Parameters.** The parameter ranges are set as follows: - For the quadratic operator, $c_0 = 0.5$, $c_1 = 1$, $c_2 = 0.1$. - For the cubic operator, $c_0 = 0.5$, $c_1 = 1$, $c_2 = 0.1$, and $c_3 = 0.01$. The dropout rate is selected from the range $[0, 0.3]$. All pooling layers are replaced with AvgPool.

**Other Hyperparameters and Settings.** The training hyperparameters include those for ResNet-18, the VanillaNet series, and ViT-Small. Additional training hyperparameters are summarized in Table 5. The loss function is defined as $\mathcal{L}_{\text{CE}} + \mathcal{L}_1 + \mathcal{L}_2$, where $\mathcal{L}_{\text{CE}}$ denotes the cross-entropy loss, and $\mathcal{L}_1, \mathcal{L}_2$ are defined in Eq. 11. Starting from the second epoch, PolyNorm is used to replace the original normalization layers during training. During inference, PolyNorm is always applied.

Table 5: Training hyperparameters for ResNet-18, VanillaNet series, and ViT-Small. Definitions of $\lambda_1$ and $\lambda_2$ are given in Eq. (11).

| Hyperparameter | ResNet-18 | VanillaNet Series | ViT-Small |
|---|---|---|---|
| Batch Size | 1600 | 960 | 300 |
| $\lambda_1$ | 0.001 | 0.001 | 0.001 |
| $\lambda_2$ | 0.01 | 0.01 | 0.01 |
| Epochs | 300 | 300 | 300 |
| Optimizer | LAMB | LAMB | LAMB |
| Learning Rate | $5 \times 10^{-3}$ | $5 \times 10^{-3}$ | $5 \times 10^{-3}$ |
| Warmup Epochs | 1 | 1 | 1 |
| PolyAct Degree | 2 | 2,3 | 3 |
| Dropout | 0.0 | 0.2 | 0.2 |
| Mixup | 0.0 | 0.2 | 0.2 |
| Cutmix | 0.0 | 1.0 | 1.0 |
| $k$ | 4 | 4 | 4 |
| $\mu$ | 2 | 2 | 2 |

## C  LLM USAGE

**Large Language Models.** We acknowledge the use of Large Language Models (LLMs) during the preparation of this paper. LLMs were employed exclusively to aid in language refinement and stylistic polishing of the manuscript. They were not used to generate research ideas, design experiments, analyze results, or mathematical derivations. All technical content, experimental design, implementation, and analysis are the sole work of the authors.

