# OpenReview forum: "ULD-Net: Enabling Ultra-Low-Degree Fully Polynomial Networks for Homomorphically Encrypted Inference"
_ICLR.cc/2026/Conference — ICLR 2026 Poster_

### Official Review · Reviewer_Lj1U · 2025-10-19

**Soundness:** 3
**Presentation:** 3
**Contribution:** 2
**Rating:** 2
**Confidence:** 4

**Summary:**

The authors propose a training framework for DNNs composed entirely of low-degree polynomial operations to make them natively suitable for privacy-preserving inference. The authors introduce PolyNorm that stabilizes training by maintaining activation ranges within well-conditioned bounds, along with a principled choice of normalization axes and auxiliary loss penalties for numerical control. ULD-Net achieves SOTA accuracy–latency trade-offs on encrypted inference, surpassing prior fully polynomial and partial polynomial methods with up to 3× faster inference and +3.3% accuracy gains, and scaling to large architectures such as ResNets and Vision Transformers.

**Strengths:**

- The work Introduces a polynomial-only normalization mechanism (PolyNorm) that effectively stabilizes ultra-low-degree polynomial networks, addressing the long-standing issue of training instability in fully polynomial architectures at ImageNet and transformer scale.
- The method demonstrates strong empirical performance and scalability than prior state-of-the-art polynomial or partially polynomial methods.

**Weaknesses:**

- Very limited novelty. The authors failed to acknowledge or compare against highly relevant and more recent prior work [1, 2, 3, 4], all of which already introduced polynomial-only architectures or co-designed low-degree activation functions for PI. These prior works have also focused on low polynomial approaches, hence the ULD-Net's "novel idea" is an incremental extensions.

- In addition to my main concern about the novelty, the proposed PolyNorm and normalization-axis heuristic are engineering refinements rather than fundamental algorithmic advances.

- The adaptive numerical constraint and normalization principles are supported by descriptive reasoning but lack formal analysis or proofs of convergence and stability guarantees. The derivations are heuristic, and there is no quantitative ablation linking PolyNorm’s theoretical formulation to observed performance gains.

- The experiments only compare against two baselines but omit many more recent and "potentially" stronger baselines [1, 2, 3, 4].

- The authors admit that the polynomial activations tend to overfit and that accuracy degrades with depth (e.g., VanillaNet-7 drops 1.6% below the original) (PolyKervNets [3] and Sisyphus [5] mention this), yet they do not propose or analyze mitigation strategies beyond vague suggestions to enhance expressive power.

**References**

[1] Park J, Kim MJ, Jung W, Ahn JH. AESPA: Accuracy preserving low-degree polynomial activation for fast private inference. arXiv preprint arXiv:2201.06699. 2022 Jan 18.

[2] Diaa A, Fenaux L, Humphries T, Dietz M, Ebrahimianghazani F, Kacsmar B, Li X, Lukas N, Mahdavi RA, Oya S, Amjadian E. Fast and private inference of deep neural networks by co-designing activation functions. In33rd USENIX Security Symposium (USENIX Security 24) 2024 (pp. 2191-2208).

[3] Aremu T, Nandakumar K. Polykervnets: Activation-free neural networks for efficient private inference. In2023 IEEE Conference on Secure and Trustworthy Machine Learning (SaTML) 2023 Feb 8 (pp. 593-604). IEEE.

[4] Ali RE, So J, Avestimehr AS. On polynomial approximations for privacy-preserving and verifiable relu networks. arXiv preprint arXiv:2011.05530. 2020 Nov 11.

[5] Garimella K, Jha NK, Reagen B. Sisyphus: A cautionary tale of using low-degree polynomial activations in privacy-preserving deep learning. arXiv preprint arXiv:2107.12342. 2021 Jul 26.

**Questions:**

- How does PolyNorm fundamentally differ from earlier polynomial stabilization methods or activation co-design approaches? What exactly does it do differently?

- Can the authors provide ablation results isolating the effect of PolyNorm and the normalization-axis principle to verify that these components are responsible for the reported stability and accuracy gains? Or can they provide a description as to what offers stability guarantees in their proposed approach.

-  Could the authors include comparisons with more recent baselines  to better contextualize ULD-Net’s performance and novelty?

---

> ### Author Response · Authors · 2025-11-24
>
> We thank the reviewer for the careful reading and detailed feedback.
>
> Our goal is to enable ultra‑low‑degree fully polynomial networks that (i) are natively compatible with homomorphic encryption (only additions and multiplications) and (ii) remain trainable and accurate at ImageNet and ViT scale. Existing works have shown promising components—low‑degree activations, polynomial approximations, or partial replacements—but, to the best of our knowledge, none has demonstrated stable ultra‑low‑degree fully polynomial training on large‑scale CNNs *and* Transformers on the ImageNet dataset.
>
> We agree that the individual pieces—polynomial activations, normalization, and HE‑friendly operators—are conceptually simple. However, in modern ML, making a known architecture robust and scalable in a new regime is itself a substantial contribution. Large language models such as GPT‑3, GPT-4 and scaling‑law work around them are widely regarded as impactful even though their advances are primarily scalability, empirical, and algorithmic rather than new convergence theory. Our work plays a similar role for fully polynomial, ultra‑low‑degree networks: we show *how* to combine simple components in a principled way so that a regime previously viewed as numerically fragile (low‑degree, fully polynomial, ImageNet/ViT scale) becomes practical.
>
> To support our claim, we
> 1. included detailed analysis the essential advantages of our work compare to prior fully polynomial networks, especially on large scale model and datasets
> 2. detailed ablation study that demonstrates the effectiveness of our method
> 3. recently trained a fully polynomial ViT-Small (ULDNet) on ImageNet, and achieved a stable **76.7%** top-1 accuracy, further demonstrating the scalability of our approach. Under the same training recipe, the reported accuracy of the original ViT-Small model is 76.5% according to report [8].
>
> Below we address each weakness and question.
>
> ---
>
> # 1.  Novelty and comparisons to prior works [1,2,3,4]
> *(Weaknesses 1, 2, 4 && Questions 1, 3)*
>
> Our work is not merely about using low-degree polynomial operators. Rather, it provides a complete recipe that makes low-degree fully polynomial networks scalable to ImageNet and ViT, while remaining stable and accurate. This recipe includes:
>
> * sample-wise normalization (with an effectiveness analysis using a simplified model),
>
> * variance-based on-the-fly normalization,
>
> * trainable polynomial coefficients,
>
> * stability-oriented loss penalty.
>
>
> Among these components, sample-wise normalization and on-the-fly scaling are the key mechanisms that enable stable and efficient training, and they represent our major **novelties** compared to prior works [1,2,3,4,5,6].
>
> It is important to clarify that prior works [1,2,3,4,5,6] are not scalable to both ImageNet and ViT. Scalability to large-scale architectures and large datasets has always been a core objective in current deep learning research, and it is especially critical—and difficult—for fully polynomial models.
>
> Most prior works [1,2,3,4,5] can train successfully on small datasets such as CIFAR but fail to scale to ImageNet according to experimental results. SMART-PAF [6], although has 81 polynomial degrees, is the strongest reproducible baseline that scales to ImageNet, and thus it is our primary point of comparison.
>
> Below we provide a detailed comparison table summarizing the key techniques and experimental results of all related works:
>
> | Work | Polynomial Degree | Polynomial Coefficient | Sample-wise Normalization | Scaling During Inference | ImageNet | ViT on ImageNet|
> | - | - | - | - | - | - | - |
> | AESPA [1]  | 2  | fixed| no| fixed  | ×  | ×  |
> | Diaa A et al. [2]| 4  | fixed| no| fixed  | 30–60%| ×  |
> | PolyKervNets [3]| 2  | trainable  | no| none  | ×  | ×  |
> | Ali R. E. et al. [4]| 2  | fixed| no| fixed  | ×  | ×  |
> | Garimella K et al. [5] | 2–9| fixed| no| fixed  | ×  | ×  |
> | SMART-PAF [6] | 81 | trainable  | no| fixed  | 69.4% (ResNet-18) | ×  |
> | Ours | 2–3| trainable  | yes  | on-the-fly| 69.8% (ResNet-18) | 76.7% (ViT-Small) |
>
> Note: Although Diaa A et al. [2] claims scalable on ImageNet, the result cannot be reproduced. Using the authors’ released code and scripts, training diverges (loss NaN) at around 120 epochs and around 60% accuracy. The original code does not replace MaxPool with AvgPool; once replaced, training collapses around 30% accuracy.
>
> *We will include a brief introduction to prior works [1,2,3,4,5] in the upcoming revision.*

---

> ### Author Response · Authors · 2025-11-24
>
> # 2.  Stability analysis and ablation studies
>
> *(Weaknesses 3 && Questions 2)*
>
> ## 2.1 Stability  analysis
>
> In Section 4.2, we derive—using a simplified model—that fully polynomial networks naturally exhibit sample-wise deviation, and that this deviation grows as the model and dataset scale up. This phenomenon is a key reason why fully polynomial models struggle with scalability. Our sample-wise normalization suppresses this deviation. In Appendix A, we show that PolyNorm contracts the variance of any input whose variance is below k times the historical mean, constraining the output variance within 1. Empirically, extreme values are very rare (Figure 2 of original paper).
>
> This is not a full mathematical proof—neither our work nor the prior works [1,2,3,4,5,6]  have the ability to provide such guarantees—but it offers a mechanism-level explanation for stable training, supported by empirical ablations.
>
> More broadly, deep learning research relies heavily on empirical and heuristic reasoning. There are limited works that can mathematically guarantee convergence or accuracy of deep neural networks on large-scale tasks.
>
> ## 2.2 Ablation studies
>
> Thank you for your suggestion. We will include the following ablation study in the appendix in the upcoming revision.
>
> Below are the first 40 epoch ImageNet training accuracy of a fully polynomial ResNet-18 using a trainable quadratic activation function, with either PolyNorm or BatchNorm (or PolyNorm with normalization axis = [B, H, W], which behaves similarly). Total epochs = 300, Batch size = 1600, learning rate = 5e-3, optimizer = LAMB.
>
> | Epoch | PolyNorm Train | PolyNorm Test | BatchNorm Train | BatchNorm Test | Epoch | PolyNorm Train | PolyNorm Test | BatchNorm Train | BatchNorm Test |
> |-------|----------------|----------------|------------------|------------------|-------|----------------|----------------|------------------|------------------|
> | 0  | 0.57 | 1.15 | 0.55 | 1.14 | 20 | 43.28 | 55.50 | 0.12 | 0.10 |
> | 1  | 1.20 | 4.74 | 1.49 | 4.50 | 21 | 43.90 | 54.30 | 0.12 | 0.10 |
> | 2  | 4.85 | 11.71 | 4.98 | 11.59 | 22 | 44.53 | 55.48 | 0.19 | 0.10 |
> | 3  | 8.42 | 16.36 | 8.82 | 16.87 | 23 | 44.92 | 57.03 | 0.10 | 0.10 |
> | 4  | 12.29 | 21.08 | 14.15 | 19.80 | 24 | 45.53 | 56.73 | 0.10 | 0.10 |
> | 5  | 16.29 | 27.87 | 17.26 | 1.48 | 25 | 46.01 | 56.89 | 0.10 | 0.10 |
> | 6  | 19.95 | 31.20 | 20.59 | 0.10 | 26 | 46.50 | 58.09 | 0.10 | 0.10 |
> | 7  | 23.11 | 33.75 | 9.99 | 0.10 | 27 | 46.87 | 57.70 | 0.11 | 0.11 |
> | 8  | 26.10 | 39.58 | 0.11 | 0.10 | 28 | 47.25 | 57.08 | 0.11 | 0.10 |
> | 9  | 28.58 | 41.11 | 0.10 | 0.10 | 29 | 47.63 | 58.94 | 0.10 | 0.10 |
> | 10 | 31.04 | 42.92 | 0.11 | 0.10 | 30 | 48.05 | 58.50 | 0.10 | 0.10 |
> | 11 | 32.95 | 46.32 | 0.10 | 0.10 | 31 | 48.42 | 58.26 | 18.82 | 12.42 |
> | 12 | 34.82 | 46.74 | 0.11 | 0.10 | 32 | 48.68 | 59.50 | 27.88 | 0.10 |
> | 13 | 36.35 | 47.73 | 7.47 | 0.10 | 33 | 49.04 | 59.38 | 27.21 | 0.11 |
> | 14 | 37.63 | 50.48 | 20.72 | 0.10 | 34 | 49.33 | 58.92 | 0.11 | 0.10 |
> | 15 | 38.85 | 50.94 | 2.02 | 0.10 | 35 | 49.62 | 60.52 | 0.11 | 0.10 |
> | 16 | 39.91 | 51.41 | 0.11 | 0.10 | 36 | 49.80 | 59.65 | 0.10 | 0.10 |
> | 17 | 40.83 | 53.37 | 0.11 | 0.10 | 37 | 50.05 | 59.45 | 0.11 | 0.10 |
> | 18 | 41.64 | 53.35 | 0.11 | 0.10 | 38 | 50.41 | 61.14 | 0.11 | 0.10 |
> | 19 | 42.49 | 53.38 | 0.12 | 0.10 | 39 | 50.63 | 60.04 | 0.11 | 0.10 |
>
> We observe that the accuracy curve of PolyNorm is smooth, whereas the BatchNorm version becomes unstable very quickly.

---

> ### Author Response · Authors · 2025-11-24
>
> The following table reports, for all 17 quadratic activation layers in the fully polynomial ResNet-18, the average input variance and input maximum at the end of several training epochs:
>
> **PolyNorm version:**
>
> | Layer | ep0 Var | ep0 Max | ep10 Var | ep10 Max | ep20 Var | ep20 Max | ep30 Var | ep30 Max |
> |-------|---------|---------|----------|----------|----------|----------|----------|----------|
> | 1  | 0.96 | 8.78 | 0.92 | 28.28 | 1.00 | 39.52 | 1.10 | 45.10 |
> | 2  | 1.47 | 10.68 | 1.26 | 54.44 | 1.22 | 95.41 | 1.20 | 103.31 |
> | 3  | 0.98 | 9.30 | 0.97 | 20.51 | 1.04 | 20.79 | 1.11 | 20.48 |
> | 4  | 2.49 | 13.47 | 1.45 | 33.52 | 1.22 | 23.01 | 1.15 | 20.88 |
> | 5  | 0.99 | 9.88 | 0.91 | 19.75 | 0.94 | 22.84 | 0.99 | 19.94 |
> | 6  | 1.82 | 13.88 | 2.30 | 29.57 | 2.30 | 49.39 | 2.30 | 53.48 |
> | 7  | 0.99 | 9.74 | 0.82 | 15.32 | 0.87 | 13.24 | 0.93 | 12.63 |
> | 8  | 2.71 | 16.68 | 1.33 | 33.59 | 1.03 | 20.00 | 0.93 | 17.40 |
> | 9  | 0.99 | 8.88 | 0.78 | 15.78 | 0.79 | 11.21 | 0.81 | 10.25 |
> | 10 | 2.15 | 13.82 | 2.03 | 21.58 | 1.94 | 25.00 | 1.89 | 24.65 |
> | 11 | 1.00 | 8.30 | 0.73 | 11.97 | 0.76 | 10.75 | 0.79 | 9.92 |
> | 12 | 3.44 | 16.20 | 0.79 | 20.37 | 0.54 | 13.49 | 0.51 | 12.28 |
> | 13 | 0.99 | 8.34 | 0.81 | 11.11 | 0.79 | 8.70 | 0.77 | 8.16 |
> | 14 | 1.82 | 10.55 | 2.36 | 16.17 | 1.98 | 17.69 | 1.72 | 15.19 |
> | 15 | 0.99 | 7.87 | 0.73 | 12.76 | 0.78 | 10.82 | 0.77 | 10.43 |
> | 16 | 2.89 | 12.70 | 0.56 | 25.33 | 0.43 | 18.26 | 0.40 | 15.69 |
> | 17 | 0.86 | 8.07 | 0.89 | 36.58 | 0.97 | 42.94 | 1.04 | 47.92 |
>
>
> **BatchNorm version:**
>
> | Layer | ep0 Var | ep0 Max | ep5 Var | ep5 Max | ep10 Var | ep10 Max | ep20 Var | ep20 Max | ep30 Var | ep30 Max | ep31 Var | ep31 Max | ep39 Var | ep39 Max |
> |--|----|---|--|---|----------|----------|----------|----------|----------|----------|----------|----------|----------|----------|
> | 1  | 1.00 | 10.64 | 1.20 | 20.35 | 1.32 | 7.80 | 1.43 | 11.04 | 1.46 | 15.10 | 1.56 | 20.79 | 1.69 | 11.17 |
> | 2  | 1.66 | 11.53 | 2.32 | 30.87 | 2.50 | 20.06 | 3.06 | 31.86 | 3.42 | 40.91 | 3.50 | 57.25 | 4.05 | 21.55 |
> | 3  | 1.00 | 10.32 | 1.33 | 27.88 | 1.48 | 10.39 | 1.65 | 16.53 | 1.85 | 39.84 | 2.02 | 19.45 | 2.25 | 23.68 |
> | 4  | 2.82 | 15.17 | 4.53 | 84.42 | 4.35 | 23.84 | 4.96 | 39.23 | 6.00 | 113.05 | 6.80 | 46.87 | 6.61 | 88.99 |
> | 5  | 1.00 | 9.98 | 1.66 | 26.03 | 1.87 | 39.80 | 2.25 | 59.83 | 2.58 | 303.10 | 2.75 | 21.95 | 2.91 | 203.91 |
> | 6  | 1.97 | 14.55 | 3.69 | 32.31 | 3.58 | 84.37 | 3.82 | 127.82 | 4.11 | 512.62 | 5.08 | 32.18 | 4.77 | 327.11 |
> | 7  | 1.00 | 10.23 | 1.66 | 29.37 | 1.79 | 128.92 | 2.03 | 200.39 | 2.16 | 412.55 | 2.42 | 30.75 | 2.63 | 424.93 |
> | 8  | 2.92 | 18.25 | 6.57 | 218.59 | 14.71 | 1499.50 | 48.18 | 1721.73 | 1512.35 | 33848.83 | 20.95 | 415.87 | 818.94 | 30430.03 |
> | 9  | 1.00 | 9.93 | 1.71 | 33.37 | 1.88 | 225.12 | 2.06 | 243.65 | 2.11 | 385.12 | 2.42 | 39.15 | 2.67 | 367.97 |
> | 10 | 2.07 | 14.35 | 3.33 | 43.88 | 3.16 | 292.04 | 3.32 | 325.28 | 3.33 | 430.04 | 3.96 | 54.82 | 3.59 | 478.70 |
> | 11 | 1.00 | 9.71 | 1.33 | 31.01 | 1.50 | 270.24 | 1.69 | 265.32 | 1.70 | 307.31 | 1.86 | 43.25 | 2.05 | 296.93 |
> | 12 | 3.25 | 18.04 | 50.73 | 1613.64 | 636.93 | 13608.99 | 916.34 | 16105.84 | 1760.45 | 30326.46 | 103.04 | 2495.63 | 2905.49 | 42610.78 |
> | 13 | 1.00 | 9.35 | 0.96 | 29.73 | 1.09 | 224.11 | 1.25 | 235.21 | 1.23 | 262.96 | 1.44 | 45.27 | 1.66 | 258.80 |
> | 14 | 2.00 | 13.65 | 1.99 | 29.32 | 0.81 | 112.01 | 0.58 | 100.83 | 0.37 | 93.37 | 1.36 | 29.74 | 0.51 | 104.65 |
> | 15 | 1.00 | 9.36 | 1.06 | 22.53 | 1.22 | 185.46 | 1.40 | 190.61 | 1.41 | 214.02 | 1.65 | 39.14 | 1.96 | 211.49 |
> | 16 | 3.23 | 18.81 | 3.14 | 178.59 | 0.32 | 134.43 | 0.09 | 67.43 | 0.03 | 48.08 | 0.97 | 43.18 | 0.08 | 73.32 |
> | 17 | 1.00 | 10.00 | 1.34 | 23.59 | 1.39 | 12.49 | 1.58 | 15.02 | 1.71 | 18.21 | 1.81 | 26.26 | 1.92 | 14.82 |
>
> We can clearly see that the PolyNorm version maintains variance at a consistently low level, enabling stable training; while the BatchNorm version experiences explosive variance growth starting around epoch 5–10, which destroys the data flow and drives accuracy to zero. This  aligns with our theoretical discussion and further demonstrates the effectiveness of our method.
>
> It also shows that the stability of fully polynomial models in real training is a very complex phenomenon. We even observe in the BatchNorm version a strange short-lived resurgence around epoch 31, where variance suddenly decreases and accuracy briefly rebounds.
>
> On the other hand, some prior works employ additional techniques that make training more stable than the BatchNorm baseline used in our ablation study. However, our method improves the stability of fully polynomial models at a more fundamental level, without relying on complex or specialized training procedures. It achieves accuracy beyond the state of the art and demonstrates strong scalability, being directly applicable to ViT architectures. We believe these results indicate that our contributions are substantial.

---

> ### Author Response · Authors · 2025-11-24
>
> # 3. On Accuracy and Expressive Power
>
> *(Weaknesses 5)*
>
> *Our latest large-scale results show that ULD-Net has almost no accuracy degradation on the ViT-Small + ImageNet scale (76.7% vs. 76.5% under the same training recipe).*
>
> We agree that in a few cases ULD-Net shows slightly lower accuracy than the original model. However, across the majority of settings, ULD-Net achieves comparable—or sometimes higher—accuracy than the corresponding baselines. Therefore, the “overfitting” behavior does not appear to be a universal or systematic limitation of ULD-Net. Our empirical results indicate that the expressive power of ultra-low-degree fully polynomial networks is largely preserved. The small accuracy fluctuations observed in individual comparisons fall within the range commonly attributed to training-time randomness. We will clarify this point in the revised version.
>
>
> ---
>
> # Summary
>
> In summary, our work achieves significant improvements over the state of the art in terms of performance, accuracy, stability, and scalability. We sincerely hope to receive your recognition. We are also ready to **open-source** this work, and we believe it will make a valuable contribution to the community.
>
> ---
>
> References:
>
> [1] Park J, Kim MJ, Jung W, Ahn JH. AESPA: Accuracy preserving low-degree polynomial activation for fast private inference. arXiv preprint arXiv:2201.06699. 2022 Jan 18.
>
> [2] Diaa A, Fenaux L, Humphries T, Dietz M, Ebrahimianghazani F, Kacsmar B, Li X, Lukas N, Mahdavi RA, Oya S, Amjadian E. Fast and private inference of deep neural networks by co-designing activation functions. In33rd USENIX Security Symposium (USENIX Security 24) 2024 (pp. 2191-2208).
>
> [3] Aremu T, Nandakumar K. Polykervnets: Activation-free neural networks for efficient private inference. In2023 IEEE Conference on Secure and Trustworthy Machine Learning (SaTML) 2023 Feb 8 (pp. 593-604). IEEE.
>
> [4] Ali RE, So J, Avestimehr AS. On polynomial approximations for privacy-preserving and verifiable relu networks. arXiv preprint arXiv:2011.05530. 2020 Nov 11.
>
> [5] Garimella K, Jha NK, Reagen B. Sisyphus: A cautionary tale of using low-degree polynomial activations in privacy-preserving deep learning. arXiv preprint arXiv:2107.12342. 2021 Jul 26.
>
> [6] Yuan G, Ju B, Chen K, Zhang Y, Chen X, Chen Y, Yang Y, Wang Y, Dai W, Wang S. Accurate low-degree polynomial approximation of non-polynomial operators for fast private inference in homomorphic encryption. In Proceedings of the 7th Conference on Machine Learning and Systems (MLSys 2024), 2024.
>
> [7] Su J, Lu Y, Pan S, Wen B, Liu Y, Wang Y, Wang S. RoFormer: Enhanced Transformer with Rotary Position Embedding. arXiv preprint arXiv:2104.09864, 2021.
>
> [8] Heo, B., Yun, S., Han, D., Chun, S., Choe, J., & Oh, S. J. (2021). Rethinking Spatial Dimensions of Vision Transformers. In Proceedings of the IEEE/CVF International Conference on Computer Vision (ICCV).

---

> > ### Comment · Reviewer_Lj1U · 2025-11-25
> >
> > The rebuttal materially improves clarity and evidential support. I now better appreciate the distinct engineering recipe that enables fully polynomial training at ImageNet/ViT scale. That said, the contribution is still primarily engineering/scaling rather than conceptual theory, and key claims need end-to-end HE measurements and standardized baselines. I still have other issues though.
> >
> > - The comparison table asserts some prior work “does not scale / is not reproducible,” including Diaa et al., USENIX’24. Please report exact configs you used (pooling replacements, quantization, schedulers, warm-ups, batch sizes) and cite commit hashes. Where fixes (e.g., Max→AvgPool) are required, show both versions. Also include SMART-PAF and PolyKervNets under identical training recipes, not just cited numbers. (Not very important)
> >
> > - Give results for longer/deeper CNNs (e.g., ResNet-50) and ViT-B/16 variants. We need to be certain about scalability to deeper layers, as those are the ones used in practice. (Very important)
> >
> > - Can you revise the current paper too, to include all the promised updates?
> >
> > Thank you and great job so far!

---

> ### Author Response · Authors · 2025-11-26
>
> Thank you for your feedback. Regarding latest questions:
>
> 1. We directly ran the code of Diaa et al. [2] using their released script without any modification:
> https://github.com/LucasFenaux/PILLAR-ESPN/blob/main/pillar/training_scripts/imagenet_resnet50.sh
>
> We recorded the console output; please see output1.txt in the supplementary material.
>
>  To replace MaxPool with AvgPool, we added the following after line 426 in train.py:
> ```
>  if "resnet" in args.model:
>     module = nn.AvgPool2d(kernel_size=3, stride=2, padding=1)
>     utils.strip(model, "maxpool", module, nn.MaxPool2d)
> ```
>  This is sufficient. We did not preserve the full console output for the AvgPool version, but we included a screenshot (snapshot_avgpool_version.png) in the supplementary material.
>
>  We will also reproduce results on SMART-PAF and PolyKerVNet later.
>
>
> 2. We are currently training fully polynomial ViT-Base on ImageNet using all available compute resource. The training is stable; we will update once we have more results.
>
>
> 3. Sure — we will update the revision as soon as possible.
>
>
> Given the additional experimental results and the clarifications we have provided—including evidence that our method achieves state-of-the-art performance—could you consider providing a more positive evaluation of our work? :)
> Thank you very much!

---

> ### Author Response · Authors · 2025-11-26
>
> We have the first-round revision and have acknowledged the prior works [1, 2, 3, 4, 5] in Section 2.2. Please see the blue-font text in Section 2.2. Thank you very much!

---

> > ### Comment · Reviewer_Lj1U · 2025-11-27
> >
> > Okay, thank you for providing the updates. As for the fully polynomial ViT-Base, I hope to see the results included and see that the method actually scales well into large models and datasets unlike other methods. Goodluck!

---

> > > ### Author Response · Authors · 2025-11-27
> > >
> > > Thank you very much!!
> > > Our ViT-Base experiment is still running, and based on the current trend, it is likely to reach 73–75%, which is consistent with the ViT variant results reported in [8] when trained purely on ImageNet-1K.
> > > As is known, ViT variants typically require pretraining on ImageNet-21K or JFT-300M to surpass 80%+ accuracy. We will continue to explore this direction in future work.
> > >
> > > Thank you again for your recognition!

---

### Official Review · Reviewer_G9h4 · 2025-11-01

**Soundness:** 3
**Presentation:** 3
**Contribution:** 3
**Rating:** 6
**Confidence:** 3

**Summary:**

To enable privacy-preserving machine learning (PPML) in homomorphic encryption (HE) settings, fully polynomial model design has become a promising approach. Previous works  suffer from high degree polynomials and high inference latency. This paper proposes normalization axis principle and two approximations of nonlinear approximation, PolyNorm (polynomial-only normalization) and PolyAct, with low-degree polynomials to enable efficient HE inference. ULD-net uses multiplicative depth less than 3 for each operators.

**Strengths:**

- The authors propose the ultra low-degree training methodology replacing the nonlinear operations in HE. Authors provide replacement of non-linear operations from previous works (Roformer) and propose PolyNorm for the normalization layer, i.e., providing full end-to-end recipe to enable HE inference.

Su, Jianlin, *et al*. "Roformer: Enhanced transformer with rotary position embedding." *Neurocomputing* 568 (2024): 127063.

- Proposed (and provides) logical mathematical reason for choosing the normalization axis. Focusing on the reduction in variance of the input for polynomial function is a logical process, and a good approach.

**Weaknesses:**

- Although ULD-Net achieved 76.40% accuracy (1.58% drop) from the largest model tested by the authors, VanillaNet-7, batch normalization and layer normalization are standard techniques, therefore the approximation (PolyNorm) might incur a performance drop on more complex task or different model.
- Formulation of the inverse square root approximation depends on $\mu$ and $k$ and the choice of them are based on experiments. $\mu$ and Var varies when dataset differs — we cannot suggest a uniform optimal choice for these statistical values.

**Questions:**

- Authors introduce approximate inverse square root function well near $\mu$, but it does not approximate well on other ranges. Why did the authors propose a new approximation of inverse square root function that have two parameters $\mu$ and $k$ instead of using the well-known approximation methods of inverse square root function such as Newton-Rapson’s method? (e.g., Cho, W. *et. al.* used Newton’s method to approximate inverse square root)

Cho, W., Hanrot, G., Kim, T., Park, M., & Stehlé, D. (2024, December). Fast and accurate homomorphic softmax evaluation. In *Proceedings of the 2024 on ACM SIGSAC Conference on Computer and Communications Security* (pp. 4391-4404).

- Determination of the best combination of parameters $(\mu,k)$ is based on experiment results without solid mathematical backbone. Will there be any mathematical reasons to determine the best parameter combination?

---

> ### Author Response · Authors · 2025-11-25
>
> We thank the reviewer for the careful reading, detailed feedback, and positive assessment of our work.
>
> As an update, ViT-Small (ULD-Net) is successfully trained on ImageNet, achieving **76.7%** top-1 accuracy. Under the same training recipe, the reported accuracy of the original ViT-Small model is 76.5% according to report [4].
> Below we address the reviewer’s questions in detail.
>
> ---
>
> # 1. Potential performance issue
>
> *(Weakness 1)*
>
> ULD-Net may not incur the drop of expressive power. As an update, our latest large-scale results show that ULD-Net has almost no accuracy degradation on the ViT-Small + ImageNet scale (76.7% vs. 76.5%). The previously reported small drop on VanillaNet-7 should be attributed to training-time randomness. We will clarify this in the revised version.
>
> # 2. PolyNorm vs. Newton iteration / Minimax polynomials
>
> *(Questions 1)*
>
> Newton iteration typically requires very high effective polynomial degree in practice.
> For example, Cho, W. et al. [1] approximates $1/\sqrt{x}$ using the Newton iteration
>
> $y_{k+1} = \frac{1}{2} y_k ( 3 - x y_k^{2} )$
>
> assuming a 4-degree minimax polynomial as $y_0$,
> the effective polynomial degrees after each iteration become:
>
> * after 0th iteration: degree 4
>
> * after 1st iteration: degree 13
>
> * after 2nd iteration: degree 40
>
> * after 3rd iteration: degree 121
>
> and so on.
> In only 3 iterations the degree already exceeds 100.
>
> Cho, W. et al. [1] does not specify how many iterations or what hyperparameters are required for real tasks.
> Correspondingly, Lee et al. [2] use minimax composite polynomials (with coefficients obtained via an improved Remez algorithm) to approximate non-polynomial operators. To make a fully-polynomial model trainable on ImageNet, Lee et al. [2] use cascaded (composite) polynomials with effective degree 6000–10000+, which is extremely high.
>  SMART-PAF [3] reduces the effective degree to 81, but relies on task-specific, sophisticated training techniques that lack generality (e.g., our attempts to train VanillaNet using SMART-PAF did not succeed).
> We have also attempted to forcibly use low-degree (4–10) Newton iteration / minimax polynomial / Chebyshev polynomial as approximations, and these settings tend to training  diverge or collapse very easily.
>
> Therefore, we adopt a fundamentally novel and more efficient approach.
> We **replace** the inverse square root with a properly designed quadratic polynomial function, rather than **approximating**  $1/\sqrt{x}$ to arbitrary precision. We prove and empirically validate that when the input variance is within a reasonable range relative to its historical running mean, PolyNorm provides sufficient on-the-fly numerical constraints (keeping the output variance $\le 1$).
> We also introduce a penalty loss to greatly reduce the probability of the variance exceeding $k$ times the running mean.
>
> Importantly, this almost does not harm model accuracy or expressive power.
> On ImageNet with ViT-Small: ULD-Net 76.7% vs. Original 76.5%.
>
> Thus, ULD-Net represents a major step forward in fully polynomial model design:
> from the previous SOTA SMART-PAF’s 81-degree ResNet-18 on ImageNet, to our 3-degree ViT-Small on ImageNet — a significant improvement in stability, scalability, and efficiency.

---

> ### Author Response · Authors · 2025-11-25
>
> # 3. On Choosing the Hyperparameters $k$ and $\mu$
>
> *(Questions 2 & Weakness 2)*
>
> We acknowledge that the optimal choices of $k$ and $\mu$ cannot currently be derived mathematically. We demonstrate empirically that choosing $k, \mu$ within a reasonable range consistently yields good results. Determining the exact optimal values for a specific architecture and dataset is difficult, which is common in deep learning.
>  For example, learning rates for optimizers are also almost always empirically selected.
>
> Real-world training involves:
>
> * large-scale architectures (~10M parameters),
>
> * large-scale datasets (~1M samples),
>
> * complex training recipes,
>
> * stochasticity at multiple levels.
>
>
> Under such conditions, empirical hyperparameter selection is reasonable and necessary.
> In all experiments in our work, we simply set $k = 4$, $\mu = 2$, and achieved strong results surpassing prior works' accuracy. We believe this is a satisfactory practical solution; exploring stronger choices remains an interesting direction for future research.
>
> ---
>
> # Summary
>
> We again thank the reviewer for the positive evaluation. We are ready to **open-source** our work, and we believe it can make a valuable contribution to the community.
>
> We hope our results — and future GitHub stars — will further demonstrate that this work deserves your recognition.
> We hope to receive your even stronger support :)
>
> ---
>
> References:
>
> [1] Cho, W., Hanrot, G., Kim, T., Park, M., & Stehlé, D. (2024, December). Fast and accurate homomorphic softmax evaluation. In Proceedings of the 2024 on ACM SIGSAC Conference on Computer and Communications Security (pp. 4391-4404).
>
> [2] Lee, J. et al. (2021). Precise approximation of convolutional neural networks for homomorphically encrypted data. arXiv:2105.10879.
>
> [3] Tong, J., Dang, J., Golder, A., Hao, C., Raychowdhury, A., & Krishna, T. (2024). Accurate Low-Degree Polynomial Approximation of Non-Polynomial Operators for Fast Private Inference in Homomorphic Encryption. In Proceedings of the Seventh Annual Conference on Machine Learning and Systems (MLSys ’24).
>
> [4] Heo, B., Yun, S., Han, D., Chun, S., Choe, J., & Oh, S. J. (2021). Rethinking Spatial Dimensions of Vision Transformers. In Proceedings of the IEEE/CVF International Conference on Computer Vision (ICCV).

---

### Official Review · Reviewer_puCa · 2025-11-02

**Soundness:** 3
**Presentation:** 3
**Contribution:** 3
**Rating:** 6
**Confidence:** 4

**Summary:**

This paper presents an interesting approach to building homomorphic encryption (HE)-friendly neural networks by introducing PolyNorm, a polynomial-only normalization technique that eliminates the non-polynomial square root operation from traditional normalization layers. The authors demonstrate solid results on small to medium-sized models, showing significant reductions in HE computational complexity (up to 2.76× speedup on ResNet-18(ImageNet)) compared to prior work, and notably extend the approach beyond CNNs to Vision Transformers, which is a positive step forward. However, the practical applicability remains questionable, as the evaluation is limited to relatively small models like ViT-Small on CIFAR and Tiny-ImageNet. At the same time, the paper lacks critical training-time measurements and any demonstration on large-scale, real-world language models—a significant gap given recent progress in applying HE to LLMs like GPT-2. More fundamentally, the core concept of designing task-specific and model-specific networks for HE is not new and dates back to early CryptoNet, which raises concerns about whether the proposed method can generalize beyond its evaluated scope, especially given that simpler polynomial-based models tend to show accuracy degradation with increased model depth (as evidenced by the 1.58% drop in VanillaNet-7). To strengthen the contribution, the authors should either demonstrate applicability to larger, more practical models, provide a detailed analysis of training overhead, or show how existing pre-trained models can be efficiently adapted to this polynomial-only framework rather than requiring complete retraining from scratch. While the technical execution is sound and the incremental improvements are noteworthy, a more transparent discussion of the method's generalizability and scalability limits would better position this work within the broader context of practical HE-based machine learning.

**Strengths:**

The paper presents a solution to a fundamental problem in HE-based neural networks—replacing the non-polynomial square root operation in normalization with a quadratic polynomial approximation (PolyNorm)—which elegantly enables training fully polynomial networks from scratch and achieves substantial HE inference speedups 2.76× to 20.5× (up to 20.5× reduction in the latency of non-polynomial operations on ViT-Small) while actually improving accuracy compared to prior post-hoc approximation methods. Beyond the technical innovation, the authors demonstrate that their approach scales beyond CNNs to Vision Transformers, breaking the traditional limitation of task- or model-specific HE designs and suggesting broader applicability across different architectures.

**Weaknesses:**

The evaluation is limited to relatively small models (ResNet-18, VanillaNet, ViT-Small on CIFAR/Tiny-ImageNet), with no evidence of scalability to larger practical models or LLMs where HE-based inference would be most valuable. Crucially, the paper provides no training time measurements or overhead analysis for ViT-Small, making it impossible to assess whether the polynomial-only design introduces significant computational burdens during training that could limit practical adoption. Furthermore, the approach still requires training models from scratch with polynomial operators rather than adapting existing pre-trained models. The already observable accuracy degradation in deeper networks (1.58% drop for VanillaNet-7) raises serious concerns about whether the method's performance will generalize to significantly larger architectures, where polynomial activation functions may struggle even more due to limited expressiveness.

**Questions:**

What are the total training times and computational costs for these models, and can the approach be scaled to train LLMs?

---

> ### Author Response · Authors · 2025-11-26
>
> We thank the reviewer for the carefully reading and the positive assessment of our work. We also have updated experimental results: ViT-Small (ULD-Net) is successfully trained on ImageNet, achieving 76.7% top-1 accuracy. Under the same training recipe, the reported accuracy of the original ViT-Small model is 76.5% according to report [1].
>
> Below we address the reviewer’s concerns.
>
> ---
>
> # On scalability to LLMs
>
> First, ULD-Net already represents a substantial advance in scalability compared to the current SOTA fully polynomial model SMART-PAF [2]: from 81-degree fully polynomial ResNet-18 on ImageNet to 3-degree fully polynomial ViT on ImageNet. This is the first fully polynomial model that scales to both a large architecture (Transformer) and a large dataset (ImageNet).
>
>  We agree that further extending scalability toward LLMs is an important future direction, and is indeed a long-term goal for fully polynomial model design. We also believe we have taken a pioneering and meaningful first step for scaling up.
>
> ---
>
> # On training cost of fully polynomial ViT-Small and the use of pretrained models
>
> We trained fully polynomial ViT-Small on ImageNet using 8×A100 with distributed data parallelism. The training requires 5–7 days, which is only 10–30% longer than training the original ViT-Small under the same setup.
>
> Moreover, for the primary application scenario of fully polynomial models—privacy-preserving inference—the main bottleneck is inference latency, not training cost. A modest increase in training time is justified if it allows fully polynomial models to significantly accelerate privacy-preserving inference.
>
> While fully polynomial models cannot directly reuse the weights of the original pretrained models, using the original pretrained models as a base for gradual polynomial replacement, as well as employing the original pretrained models as teachers for distillation training, holds great potential for reducing the required training epochs. Thank you for the suggestion and we will explore this direction in future work.
>
> ---
>
> # On accuracy & Expressive
>
> Our latest large-scale results show that ULD-Net exhibits almost no accuracy degradation at the ViT-Small + ImageNet scale (76.7% vs. 76.5%). The previously reported small drop on VanillaNet-7 should be attributed to training-time randomness. We will clarify this in the revised version.
>
> ---
>
> # Summary
>
> We again thank the reviewer for the positive evaluation. We are ready to **open-source** our work and believe it will make a valuable contribution to the community.
>
> We hope our updated results — and the future GitHub stars — will further demonstrate that the work deserves your recognition.
> We sincerely hope to receive your strong support.
>
> ---
>
> References:
>
> [1] Heo, B., Yun, S., Han, D., Chun, S., Choe, J., & Oh, S. J. (2021). Rethinking Spatial Dimensions of Vision Transformers. In Proceedings of the IEEE/CVF International Conference on Computer Vision (ICCV).
>
> [2] Tong, J., Dang, J., Golder, A., Hao, C., Raychowdhury, A., & Krishna, T. (2024). Accurate Low-Degree Polynomial Approximation of Non-Polynomial Operators for Fast Private Inference in Homomorphic Encryption. In Proceedings of the Seventh Annual Conference on Machine Learning and Systems (MLSys ’24).

---

> > ### Comment · Reviewer_puCa · 2025-11-27
> >
> > The authors' response clarifies two important points. First, I agree that my original wording ("entire experiment setup uses artificially sparse secrets") was too strong: the paper does include experiments on dense secrets, and the proposed method is in principle applicable to unconstrained CBD‑type distributions. However, for Kyber/ML‑KEM‑like parameter regimes, the main empirical results still rely on sparse secrets, and the dense‑secret experiments remain in significantly lower‑dimensional or non‑standard settings. Thus my conclusion that the work does not directly threaten standardized ML‑KEM remains unchanged.
> > Second, I appreciate the clarification about the advantages of a white‑box robust linear model compared to SALSA‑style transformers. This does strengthen the interpretability/efficiency angle compared to my initial reading. That said, the learning component still uses off‑the‑shelf robust regression without new algorithms or theoretical insights, so I continue to view the main contribution as cryptanalytic/algorithm‑engineering rather than machine‑learning‑methodological.
> > Finally, the authors’ complexity estimates for real ML‑KEM parameters show that their NoMod attack is strictly more expensive than existing classical lattice attacks. This confirms that the work is best interpreted as a tool for exploring security margins under relaxed assumptions, rather than as a practical break of standardized schemes.
> > Overall, while the rebuttal helpfully corrects some overstatements in my review and clarifies the scope, it does not change my assessment regarding ML novelty or venue fit. I therefore keep my rating.

---

> > > ### Author Response · Authors · 2025-11-27
> > >
> > > Dear reviewer, may I confirm whether this comment is referring to our work?

---

### Official Review · Reviewer_2kfr · 2025-11-08

**Soundness:** 3
**Presentation:** 3
**Contribution:** 3
**Rating:** 6
**Confidence:** 4

**Summary:**

The main bottleneck in performing homomorphic computation on modern AI models such as ResNet and ViT lies in the non-linear operations and non-linear normalization layers. These operations are known to play a crucial role in the overall performance of these models.
In this paper, we propose a method to replace all non-linear operations in these models with polynomial operations while maintaining comparable performance.

**Strengths:**

The non-polynomial parts of LayerNorm are replaced with a quadratic function and ensured numerical stability by fixing the normalization axis. Additionally, softmax attention and ReLU/GELU functions are replaced with low-degree polynomials, while still demonstrating strong performance on ResNet and ViT models — thereby validating the practical effectiveness of their approach.

Although there have been several prior attempts to approximate non-polynomial functions with low-degree polynomials, the introduction of normalization-axis fixing in LayerNorm to improve training stability stands out as a clear and meaningful contribution. One potential concern is whether this simplification might lead to a loss of expressiveness or overfitting issues, but the authors address this through penalty regularization terms and dropout, successfully mitigating such risks. Their experiments on the ImageNet dataset show accuracy comparable to existing methods, which effectively demonstrates that the model’s representational capacity is preserved.

This finding has significant implications for privacy-preserving machine learning (PPML): it suggests that by incorporating axis fixing, penalty loss terms, and dropout during training, one can approximate non-polynomial layers using only low-degree polynomial functions without severe performance degradation. Even though the paper does not explicitly explore this, the same methodology could likely be extended to other non-polynomial layers.

Finally, the paper clearly describes the conversion procedure for non-polynomial layers, which makes the proposed approach extensible and comparable for future applications in other architectures.

**Weaknesses:**

There was no comparison with state-of-the-art (SOTA) models for ViT in the paper. Among the current SOTA transformer models implemented under homomorphic encryption, two representative examples are THOR and PowerFormer. As far as I know, the NEXUS model is less efficient compared to these two models. Therefore, in Table 3, for the comparison of Softmax, LayerNorm, and GELU, the authors should include THOR and PowerFormer rather than limiting the comparison to NEXUS.

Moon, Jungho, et al. "THOR: Secure transformer inference with homomorphic encryption." CCS 2025.
Park, Dongjin, et al. "Powerformer: Efficient and High-Accuracy Privacy-Preserving Language Model with Homomorphic Encryption." ACL 2025.

**Questions:**

I would like to know how the convolution layer was implemented. There are two possible methods — the MPConv approach and the Neujeans approach. Since this paper does not use bootstrapping (as it is implemented using SEAL), I assume it employs the MPConv method. Is that correct?

If bootstrapping were to be applied, the Neujeans method could also be used. Do you think incorporating this method would further improve performance?

Also, for the ViT comparison, it would be good to include THOR and PowerFormer models in the evaluation.

---

> ### Author Response · Authors · 2025-11-26
> **Reply to Reviewer 2kfr**
>
> We thank the reviewer for the careful reading of our manuscript and the constructive comments and suggestions, and we are also grateful for the positive evaluation of our work.
>
> To clarify our latency evaluation setup, we use the Multiplexed-Convolution [1] implementation with bootstrapping enabled to measure end-to-end inference latency. NeuJeans[5] convolution design performs a slot-to-coefficient transform inside the bootstrapping procedure, effectively fusing the convolution with bootstrapping. As a result, each bootstrapping operation can accommodate exactly one convolution layer.
>
> For example, in ResNet-18 evaluation with ULD, it uses two bootstrapping operations over the entire inference pipeline. Under NeuJeans’ design, this means that at most two convolution layers can benefit from their convolution-with-bootstrapping optimization in our setting. We therefore apply their fused convolution to two convolution layers, and the resulting performance speedups are reported as follows.
>
> | Benchmark    | W/O   | W     |
> |--------------|-------|-------|
> | ResNet18     | 41408 | 39165 |
> | VanillaNet-5 | 3469  | 3298  |
>
> Again, we thank for the reviewer’s suggestion to consider other SOTA methods, specifically THOR[3] and PowerFormer[4]. We have implemented their proposed nonlinear function approximations and incorporated them into our evaluation pipeline. The corresponding comparison results for each non-linear components are reported in the attached tables.
>
> **Softmax**
> | ViT          | Ours | THOR  | PowerFormer | NEXUS |
> |--------------|------|-------|-------------|-------|
> | CIFAR10      | 156  | 6522  | 315         | 3055  |
> | TinyImageNet | 472  | 19765 | 955         | 9259  |
>
> **LayerNorm**
> | ViT          | Ours | THOR  | PowerFormer | NEXUS |
> |--------------|------|-------|-------------|-------|
> | CIFAR10      | 156  | 4545  | 236         | 2080  |
> | TinyImageNet | 474  | 13804 | 716         | 6304  |
>
> **GELU**
> | ViT          | Ours | THOR | PowerFormer | NEXUS |
> |--------------|------|------|-------------|-------|
> | CIFAR10      | 78   | 2154 | 609         | 2860  |
> | TinyImageNet | 236  | 6529 | 1844        | 8668  |
>
> We will incorporate the new comparison results into the upcoming revision, and we once again thank the reviewer for the recognition of our work. We are ready to open source our implementation, which we believe will be valuable to the community. We hope that the experimental results and the community’s reception of our open source release will further demonstrate that this work deserves your recognition, and we would greatly appreciate your continued strong support.
>
> Reference:
>
> [1] Multiplexed: Lee, Eunsang, et al. "Low-complexity deep convolutional neural networks on fully homomorphic encryption using multiplexed parallel convolutions." International Conference on Machine Learning. PMLR, 2022.
>
> [2] NEXUS: Zhang, Jiawen, et al. "Secure Transformer Inference Made Non-interactive." NDSS. 2025.
>
> [3] THOR: Moon, Jungho, et al. "THOR: Secure transformer inference with homomorphic encryption." CCS  (2025).
>
> [4] PowerFormer: Park, Dongjin, Eunsang Lee, and Joon-Woo Lee. "Powerformer: Efficient and High-Accuracy Privacy-Preserving Language Model with Homomorphic Encryption." Proceedings of the 63rd Annual Meeting of the Association for Computational Linguistics (Volume 1: Long Papers). 2025.
>
> [5] NeuJeans: Ju, Jae Hyung, et al. "Neujeans: Private neural network inference with joint optimization of convolution and FHE bootstrapping." Proceedings of the 2024 on ACM SIGSAC Conference on Computer and Communications Security. 2024.

---

### Author Response · Authors · 2025-11-30
**Final Remark & Appreciation**

## Dear Area Chair, we would like to note that our score was updated to 6, 6, 6, 6 on 27 Nov 2025 at 03:26 EST, well before the leak incident escalated. We sincerely thank all reviewers for their valuable feedback and positive evaluations of our work.

---

# Summary:

In the initial review stage, Reviewers G9h4, puCa, and 2kfr expressed consistently positive assessments of our work and each assigned a score of 6.

Reviewer Lj1U initially gave a score of 2, raising questions about novelty & scalability and requesting a more detailed comparison with several prior works, as well as ablation studies to demonstrate the effectiveness of our method.

We provided comprehensive responses to all reviewers, and in particular to Reviewer Lj1U:

* we included detailed comparisons with all relevant prior works to clearly articulate our innovations;

* we presented performance summaries showing that our method surpasses all prior works in a large margin;

* we provided extensive additional ablation studies that clearly verify the effectiveness of our approach;

* and we further supplied extra promising experimental results demonstrating that our method scales to larger model architectures (ViT) and datasets.

Our responses directly addressed all concerns raised by Reviewer Lj1U and demonstrated that our work substantially advances beyond prior efforts.

**Reviewer Lj1U acknowledged our clarifications, agreed with our contributions, and updated their score to 6 on 27 Nov 2025 at 03:26 EST. This can be verified in the [Revision History](https://openreview.net/revisions?id=HLXyxYM9ie).**
As a result, our final score became 6, 6, 6, 6.

We sincerely hope that the Area Chair will evaluate this submission fairly and reasonably, based on the objective and positive assessments formed jointly by the reviewers and us after thorough academic discussion.

In summary, our work achieves significant improvements over the state of the art in performance, accuracy, stability, and scalability. We are grateful for the constructive feedback and positive evaluations from the reviewers.
We are also fully prepared to **open-source this work**, and we believe it will make a valuable contribution to the community, and demonstrate that this work truly merits the recognition of the reviewers and the Area Chair.

Thank you very much!

---

Update: We would like to share some good news. Our fully polynomial ViT-Base on ImageNet has achieved 75.2% top-1 accuracy (original 75.3% [1]).
Together with our earlier result — fully polynomial ViT-Small reaches 76.7% (original 76.5% [1]) — this marks a significant milestone: we have successfully scaled fully polynomial model design to the ViT/ImageNet level for the first time.


---


Reference:

[1] Heo, B., Yun, S., Han, D., Chun, S., Choe, J., & Oh, S. J. (2021). Rethinking Spatial Dimensions of Vision Transformers. In Proceedings of the IEEE/CVF International Conference on Computer Vision (ICCV).

---

### Meta-Review · Area_Chair_MtDx · 2026-01-08

**Summary:**

During the initial review stage, Reviewers G9h4, puCa, and 2kfr each provided positive evaluations of the submission and assigned scores of 6, indicating that they found the work technically sound, well-motivated, and of interest to the community. In contrast, Reviewer Lj1U initially raised concerns regarding the novelty of the proposed approach, its scalability, and the need for more comprehensive comparisons with prior work and ablation studies, resulting in a lower initial score (2).

Following the rebuttal and discussion period, Reviewer Lj1U acknowledged the authors’ clarifications and updated their assessment, with the discussion converging on clarifying the paper’s novelty, empirical validation, and scalability relative to existing approaches. Overall, the rebuttal effectively addressed many of the reviewer’s concerns, and the paper presents a valuable and timely perspective on engineering polynomial networks under practical system constraints.

That said, several **suggestions** could further strengthen the final version of the paper:

* **Scalability discussion.** While the proposed low-degree polynomial design is effective for the evaluated settings, it would be beneficial to more explicitly discuss how this approach scales to larger neural networks. In particular, clarifying whether low-degree polynomials remain effective at scale—or whether higher degrees become necessary, introducing new trade-offs—would help contextualize the applicability of the method.

* **Broader experimental validation.** The experimental evaluation could be broadened beyond vision models. Adding results on text-based language models (e.g., BERT or GPT-2) would strengthen the claim that the approach generalizes across modalities, rather than being specialized to vision architectures.

* **Reproducibility and implementation details.** While the added ViT classification results are helpful, providing more detailed implementation descriptions and releasing reproducible code would significantly improve the paper’s clarity and impact.

* **Clarification of novelty.** The novelty would benefit from sharper articulation, particularly by explicitly separating and highlighting contributions from both the machine learning and FHE perspectives, and positioning them more clearly relative to prior work. This would help readers better understand how the proposed techniques advance the state of the art beyond existing polynomial or encrypted inference methods.

In summary, the paper offers a solid and promising contribution, especially from an engineering and systems perspective. With clearer articulation of novelty, broader empirical validation, and an expanded discussion of scalability trade-offs, the work would be well-positioned for acceptance and strong impact in the final version.

**Reviewer Concerns:**

The rebuttal directly addressed the primary concerns raised by Reviewer Lj1U. In particular, the authors provided more detailed comparisons with relevant prior work, additional ablation studies, and further experimental results demonstrating scalability to larger vision models and datasets. These responses were acknowledged by Reviewer Lj1U, who indicated that their concerns had been partially addressed and revised their score accordingly. The reviewer also suggested discussing the attention-based Transformer in vision Transformers. Although the authors added results for ViT on classification tasks, the implementation details and reproducible code are not available for evaluation. Importantly, it is not clear whether it will be extended to language Transformers. It seems not easy to scale up using the proposed low-degree polynomial. In large neural networks, it will no longer be low-degree but high-degree, which is a trade-off. However, there is no such discussion.

**Reviewer Scores:**

Reviewer G9h4: Assigned an initial score of 6. With full participation, this reviewer would likely have maintained a similar score, as no major concerns were raised.

Reviewer puCa: Assigned a score of 6 and expressed a generally positive assessment. Full participation would likely have confirmed this evaluation.

Reviewer 2kfr: Assigned a score of 6 and did not identify critical issues. Their score would likely have remained stable with full discussion.

Reviewer Lj1U: Initially assigned a 2, but after reviewing the rebuttal and engaging with the discussion, updated their score. The novelty and challenges of the proposed solution are only partially addressed.

---

### Decision · Program_Chairs · 2026-01-26

Accept (Poster)